# A Multi-Global Navigation Satellite System (GNSS) Time Transfer Method with Federated Kalman Filter (FKF)

**DOI:** 10.3390/s23115328

**Published:** 2023-06-04

**Authors:** Kun Liang, Shuangyu Hao, Zhiqiang Yang, Jian Wang

**Affiliations:** 1School of Electronics and Information Engineering, Beijing Jiaotong University (BJTU), Beijing 100044, China; 2National Institute of Metrology (NIM), Beijing 100029, China

**Keywords:** GNSS, time transfer, multi-GNSS, data fusion

## Abstract

Relative to single Global Navigation Satellite System (GNSS) measurements, i.e., the measurements from a single GNSS system, a single GNSS code, and a single GNSS receiver, multi-GNSS measurements for time transfer could improve reliability and provide better short-term stability. Previous studies applied equal weighting to different GNSS systems or different GNSS time transfer receivers, which, to some extent, revealed the improvement in the additional short-term stability from the combination of two or more kinds of GNSS measurements. In this study, the effects of the different weight allocation for multi-measurements of GNSS time transfer were analyzed, and a federated Kalman filter was designed and applied to fuse multi-GNSS measurements combined with the standard-deviation-allocated weight. Tests with real data showed that the proposed approach can reduce the noise level to well below about 250 ps for short averaging times.

## 1. Introduction

Allan and Weiss demonstrated that the Global Navigation Satellite System (GNSS) time transfer can provide good accuracy and precision over different baselines [1]. In International Atomic Time (TAI) generation, time transfer by the Global Positioning System (GPS) has especially been employed regularly for remote comparison links between the Physikalisch-Technische Bundesanstalt (PTB), Braunschweig, Germany, and other TAI laboratories. Version 2E of the Common Generic GNSS Time Transfer standard (CGGTTS) [2] provides a method to employ the data from the Global Positioning System (GPS), the BeiDou Navigation Satellite System (BDS), the European Galileo, and the GLObal NAvigation Satellite System (GLONASS) for remote time and frequency transfer. The time transfer experiments using the equipment via different GNSS systems, especially the BDS and Galileo systems on the different links, have been implemented for the evaluation on the long-term stability and accuracy by Kong et al. [3], Huang and Defraigne [4], Liang et al. [5,6,7], and Zhang et al. [8].

The data fusion with weight allocation method is widely used to imporove the accuracy and stability of the solution. Farinaz et al. used weighted least-squares (WLS) to calculate the tropospheric delay to improve positional accuracy [9]. Zhou et al. provided more precise positioning results with the iGMAS multi-GNSS combined orbit and clock products [10]. Yin et al. presented a more robust and more accurate localization by fusing GNSS and IMU Data with a smoothed-error-state Kalman filter [11]. For the fusion of multi-GNSS links, Jiang and Lewandowski provided the results with the simple unweighted average combination of GPS and GLONASS measurements epoch by epoch between the All-Russian Scientific Research Institute of Physicotechnical and Radio Engineering Measurements (VNIIFTRI), Mendeleevo, Moscow region, Russia and the PTB, and then accordingly noted the improvements in noise level, stability, and robustness [12]. Harmegnies et al. presented the combination of GPS and GLONASS into one unique time transfer solution [13]. The noise level of GPS and GLONASS ionosphere-free combinations in all-in-view (AV) solutions were compared to the GPS-only all-in-view solutions. Liang et al. took the simple mean value of GPS P3 code (the ionosphere-free combination of P1 and P2 codes) and BDS P3 code (the ionosphere-free combination of C1 and P2 codes) [14]. From the results, better robustness and short-term stability were shown with the combined method. Lin and Jiang proposed a virtual GPS receiver based on a weighted average of multiple calibrated receivers [15,16]. The experimental results for long baselines showed that multi-receiver fusion had improved robustness and short-term stability. Equal weighting is commonly used in these references. However, this could not be the optimal method for combining multiple GNSS time links.

For multi-GNSS time transfer, we compared the data fusion effects among the enumerated different weights, standard deviation (std)-allocated weights, and time deviation (TDEV)-allocated weights. The federated Kalman filter (FKF) was proposed for fusing the multi-GNSS time measurement, and it was used to implement the experiments. The results showed that the use of a federated Kalman filter based on std-allocated weight could reduce the noise level of the time transfer and further improve the additional short-term stability to better than 250 ps and 117 ps, respectively. Certainly, it will also increase the redundancy and improve the reliability for time transfer.

## 2. Materials and Methods

### 2.1. GNSS Time Transfer

The basic structure for GNSS time transfer is shown in Figure 1. GNSS time transfer receivers R1 and R2 are referenced to the corresponding local time references LTR1 and LTR2 at the two sites. Each receiver obtains the time difference between the local time reference and the GNSS time and the difference between LTR1 and LTR2 is calculated.

Common clock difference (CCD) experiments can be implemented to characterize the performance of the noise level or the additional short-term stability for the link. CCD experiments were used to evaluate the noise level between different satellites and receiver links (see [5]) and to verify the capacity of the NIM-TF-GNSS-3 receiver and RinCGG software for BDS time transfer (see [6]). The short-term stability of time links was evaluated by CCD experiments at NIM in Beijing (see [7]). Remote time transfer experiments are implemented to characterize the accuracy and the stability in the longer term.

### 2.2. Weighted Data Fusion with Different Weights

To find the effects for a different weight allocation, the different weight combination was enumerated with a step of 0.1 in two-link fusion and in three-link fusion and the std values were observed. Moreover, the std and TDEV in 960 s of the GNSS measurements were used as the parameters to allocate the weight. Here, the combination of multiple GNSS systems was made as an example, and the weight allocation principles are as shown in (1) and (2).
(1)$$\lambda =-\frac{2}{\sum_{i=1}^{n}\frac{1}{\sigma_{i}^{2}}}$$
(2)$$w_{i}=-\lambda /2\sigma_{i}^{2}$$

$\sigma_{i}$ is the parameter that shows the precision for the measurement from the *i*-th GNSS system, e.g., std; $w_{i}$ is the weight of the *i*-th GNSS system. Formulas (1) and (2) are used in the later fusion experiments with std-allocated weights.

The experiments were implemented for the fusion of two CCD time links with GPS and BDS systems and the fusion of three CCD time links with GPS, BDS, and GLONASS systems. All the weighted fusion results were compared to the ones with the single GNSS system. All the CCD GNSS time links used for the analysis were between IM15 and IM21 (both types: NIM-TF-GNSS-3) stations during Modified Julian Date (MJD) 59,027–MJD 59,056. Both stations were referenced to UTC(NIM) at the Changping campus of the National Institute of Metrology (NIM), Beijing, China. In [5,6], the experiments were implemented involving NIM receivers (type NIM-TF-GNSS-3), Dicom GTR receivers (type GTR series), and Septentrio receivers (type PolaRx series). The noise level (i.e., std) was evaluated as about 1 ns by the CCD results.

Table 1 and Table 2 show the std of the fusion of CCD results, which come from GNSS time links during MJD 59027–MJD 59056. For two-link fusion, the std value of the results with std-allocated weight is 0.3117 ns; at this time, the weights of GPS and BDS systems are, respectively, 0.3665 and 0.6335. For our enumeration with a step of 0.1 in Table 1, the best weight combination 0.4 and 0.6 receives the smallest std value, which is quite near the std-allocated weights. The same conclusion can be found in Figure 2 with a step of 0.02, and the std values with std-allocated weight reach the lowest point. Compared to the best single-system result, the result with the std-allocated weight method is improved by 19.8%. Compared to the equal weighted averaging, the result with the std-allocated weight method is improved by 3.4%. However, this is not absolute, and we could infer that the improvement can be greater if the performance of two systems differs more.

For the three-link fusion, the best std still comes from the std-allocated weight, and the corresponding weight combinations for GPS, BDS, and GLONASS are 0.3412, 0.5896, and 0.0692, which is similar to what Figure 3 shows with a step of 0.1. In the enumeration, the nearest weight combination is 0.4000, 0.6000, and 0 in Table 2 with a step of 0.2. Compared to the best single-system result, the result with the std-allocated weight method is improved by 22.0%.

From all the results, the std value for the corresponding fusion results is optimal when weight allocation fusion via std is used in the multi-GNSS time transfer.

### 2.3. Multi-GNSS Time Transfer

The federated Kalman filter, as shown in Figure 4, is one of the main methods to realize multi-sensor information fusion [17], since it has the advantages of mainly noise mitigation, good redundancy, and fault tolerance from the fused results. FKF is able to share information among data sources or sensors to improve the accuracy of estimates. Each sensor of FKF is equipped with its own Kalman filter, state, and observed variables. When there are different input measurements, the federated Kalman filter weights each contribution with $\beta_{i}$ by calculating the measurement covariance matrix $P_{i}$ for the local filter i, as shown below. $\beta_{i}$ could be allocated with the std-weighted method. At the same time, the local filtering is implemented on each measurement input. In Figure 4, data1, data2, and data3 represent the results of BDS B1I, GPS C1, and GLONASS C1 code measurements in CCD experiments, or the results of BDS L3B, GPS P3, and GLONASS P3 code measurements in remote time transfer experiments, respectively.

With the FKF, the multi-GNSS time transfer method could be as follows. In the local filter, the state vector X(k) at the time epoch k contains three variables, which are the clock difference x(k), the frequency offset b(k), and the aging coefficient c(k). The state vector X(k) can be expressed as
(3)$$X(k)={\left[x(k)\;b(k)\;c(k)\right]}^{T}$$

The recursive process of the system from the time epoch k−1 to k is expressed as
(4)X(k)=Φ(k−1)X(k−1)+w(k−1)
where $w(k)={\left[q_{1},q_{2},q_{3}\right]}^{T}$ is the noise input to the system. $q_{1}$, $q_{2}$, and $q_{3}$ could be the variances in white noise frequency, random walk noise frequency, and random running noise frequency involved in the time difference data, respectively. The state propagation matrix Φ(k) and the measurement matrix H(k) are expressed as
(5)$$\Phi (k)=\left[\begin{array}{ccc} 1 & \tau & \tau^{2}/2\\ 0 & 1 & \tau \\ 0 & 0 & 1 \end{array}\right],\;H(k)=\left[\begin{array}{ccc} 1 & 0 & 0 \end{array}\right]$$
where τ is the sampling interval. The measurement vector Z(k) is expressed as (6) and V(k) is the measurement noise.
(6)Z(k)=H(k)X(k)+V(k)

Since the error of the state vector after the *k*-th measurement is X(k|k)−X(k), the variance matrices of the errors are described as
(7)$$P(k|k)=E\{\;\left[\;\left|X(k|k)-X(k)\right|\;\right]{\left[\;\left|X(k|k)-X(k)\right|\;\right]}^{T}\;\}$$
(8)$$P(k|k-1)=E\{\;\left[\;\left|X(k|k-1)-X(k)\right|\;\right]{\left[\;\left|X(k|k-1)-X(k)\right|\;\right]}^{T}\;\}$$

The measurement covariance matrix of the measurement noise R(k) and the covariance matrix of the state noise Q(k) are defined as
(9)$$R(k)=E[V(k)V{\left(k\right)}^{T}]=\left[q_{4}\right]$$
(10)$$Q(k)=E[w(k)w{\left(k\right)}^{T}]$$
where $q_{4}$ is the variance of the measurement noise, and the measurement noise can be assumed as white noise following a normal distribution with zero mean. Fan et al. demonstrated that the process covariance matrix Q(k) can be expressed as
(11)$$Q(k)=\left[\begin{array}{ccc} q_{1}\tau +q_{2}\tau^{3}/3+q_{3}\tau^{5}/20 & q_{2}\tau^{2}/2+q_{3}\tau^{4}/8 & q_{3}\tau^{3}/6\\ q_{2}\tau^{2}/2+q_{3}\tau^{4}/8 & q_{2}\tau +q_{3}\tau^{3}/3 & q_{3}\tau^{2}/2\\ q_{3}\tau^{3}/6 & q_{3}\tau^{2}/2 & q_{3}\tau \end{array}\right]$$
where $q_{1}$, $q_{2}$, and $q_{3}$ are obtained through noise analysis of time difference data to complete the local filtering [18]. The corresponding state estimate $X_{i}$ obtained from the local filter i is globally fused together with the state estimate $X_{m}$ from the master filter *m*, which is described as follows.
(12)$${\sum \beta }_{i}+\beta_{m}=1$$
(13)$$P_{f}={\left(\sum P_{i}^{-1}+P_{m}^{-1}\right)}^{-1}$$
(14)$$P_{i}=P_{f}\beta_{i}$$
(15)$$P_{m}=P_{f}\beta_{m}$$

Then, the corresponding information distribution coefficient $\beta_{i}$ and $\beta_{m}$ for the local filter and the master filter will be allocated, the final estimate $P_{f}$ will be acquired, and $X_{f}$ will be generated with weighted fusion. These results are fed back to the local filter to correct the system itself so that the state of the Kalman filter in the process tends to be zero and then the closed loop of the federated Kalman filter is realized.

## 3. Results

For verification of multi-GNSS time transfer with FKF, the CCD and the remote time transfer experiments were implemented. The code measurements via the GPS, BDS, and GLONASS were involved, and both data from single-frequency measurements with the ionospheric delay compensated by the Klobuchar model and dual-frequency measurements were employed. The CCD measurements were processed through the GNSS common-view (CV) method and the remote time transfer measurements were processed through the all-in-view (AV) method. The time link measurements and the corresponding TDEV results were employed for analysis.

All the receivers in the experiments were referenced to one H-maser or a time scale based on one H-maser or one cesium clock. The CCD experiments were implemented to investigate the relation between the additional frequency stability of the time transfer link and the stability of the H-maser or the cesium clock. From the Allan deviation (ADEV) in Figure 5, at the averaging times of around 10,000 s, GNSS code signals become more stable than cesium clocks, and GNSS carrier phase (CP) signals become more stable than the H-maser. That is to say, generally, when the averaging time exceeds tens of thousands of seconds, the GPS carrier phase link can characterize the H-maser and the GNSS code link can characterize the cesium clock.

Multi-GNSS combines the measurements from multiple GNSS systems, which increases the number of satellites, achieves a better DOP, and gains more measurements.

The CCD experiments via BDS B1I, GPS C1, and GLONASS C1 code measurements were implemented between the receivers IM15 and IM21 during MJD 59027–MJD 59056, and both were referenced to UTC(NIM). In Figure 6, std values of the fusion results using the std-weighted method and the unweighted average method, respectively, were 0.30 ns and 0.44 ns, and std for the best single GNSS measurements was 0.39 ns. Using a Kalman filter with parameters *R* = 1 and *Q* = 1 to filter the fusion results of the unweighted average method, std was reduced to 0.27 ns. For the FKF time transfer link, std was reduced to 0.25 ns with the combination of fusion and filtering (*R* = 1 and *Q* = 1), and was improved by 43.2% and by 35.9%, respectively, compared to the unweighted average results and BDS results. Figure 7 shows the additional time stability of the FKF link, i.e., TDEV was the best and better than 117 ps. When the averaging time was 960 s, the time stability of the FKF link results was significantly improved; the time stability of the FKF link results improved by 52.2% compared to that of the BDS results, was 44.8% better than that of the unweighted average results, and was 10.8% better than that of the Kalman filtering results. The FKF time transfer link with filtering and fusion not only had better redundancy and reliability but also acquired the least noisy effects, as shown in Figure 6.

Via BDS, GPS, and GLONASS, the remote time transfer experiments were implemented between the receivers IM21 and IM15 during MJD 57,920–MJD 57,949. IM21 was referenced to UTC(NIM) and IM15 was referenced to a cesium clock (type symmetricom 5071A) with the high-performance tube at the International Bureau of Weights and Measures (BIPM), Sevres, France. The baseline length between IM15 and IM21 was about 7656 km. BDS L3B, GPS P3, and GLONASS P3 code measurements were selected. A Kalman filter with parameters *R* = 1 and *Q* = 1 was used to filter the unweighted average results. Since the satellite coverage for the BDS-2 system was quite poor at the BIPM site, relative to that at NIM (see details in [5,6]), std for weight allocation for BDS was taken as 2.36 ns, i.e., the std of CCD was between two NIM-TF-GNSS-3 receivers at BIPM, from [6] at the worst, and the values for GPS and GLONASS were 0.51 ns and 1.13 ns from Table 2, respectively. It could be seen from Figure 8 that the results of the FKF link agree well in terms of trend with those of the other links. From TDEV plots in Figure 9, the additional time stability by the FKF time transfer method was improved in the short term compared to the results with the single GNSS system. When the averaging time was 960 s, the time stability of the FKF link results improved by 29.8% compared to that of the GPS results, 60.9% better than that of the unweighted average results, and 37.2% better than that of the Kalman filtering results. In the short term, the TDEV plots with the unfiltered GPS CP results by NRCan_PPP software with IGS final products were closest to those of the cesium clock. Starting from an averaging time roughly beyond 100,000 s, the time stability with the different time transfer methods tended to be similar. The TDEV plots with the FKF link results were closest to those of the unfiltered GPS CP results.

The CCD and the long baseline with 7656 km length experiments characterized the typical performance of GNSS time transfer links, such as noise level and accuracy. The results showed that the additional time stability of the FKF link for the three GNSS systems was better than that of the single GNSS system with code measurements. Longer baselines introduce more link noise. Therefore, in principle, the FKF time link can achieve similar performance when the baseline length is shorter than 7656 km.

## 4. Conclusions

In this work, multi-GNSS measurements were used for time transfer. Three weight-allocated methods were compared, and the std-allocated weight method led to the best data fusion results. Compared to the best single-system result, the std value for the results of two-link fusion and three-link fusion with std-allocated weight improved by 19.8% and 22.0%, respectively. The FKF time transfer with std-allocated weight for multiple GNSS systems was used to implement the time transfer experiments from the multi-GNSS system. The FKF time links combining the multi-GNSS measurements were generally improved in stability in the short term compared to the original time link with the single GNSS measurements. In CCD experiments, the noise level (i.e., std) for the FKF time link was reduced by 43.2% and 35.9%, respectively, compared to the unweighted average results and the best single-GNSS measurements; the additional time stability for the FKF time link was improved to better than 117 ps, and the short-term time stability at 960 s was improved by 44.8% and 52.2%, respectively, compared to the unweighted average results and the best single-GNSS measurements. The long baseline results gave the similar effects. The results have shown that the FKF time transfer for multi-GNSS measurements can bring the effective improvements to reliability and noise level.

## Figures and Tables

**Figure 1 sensors-23-05328-f001:**
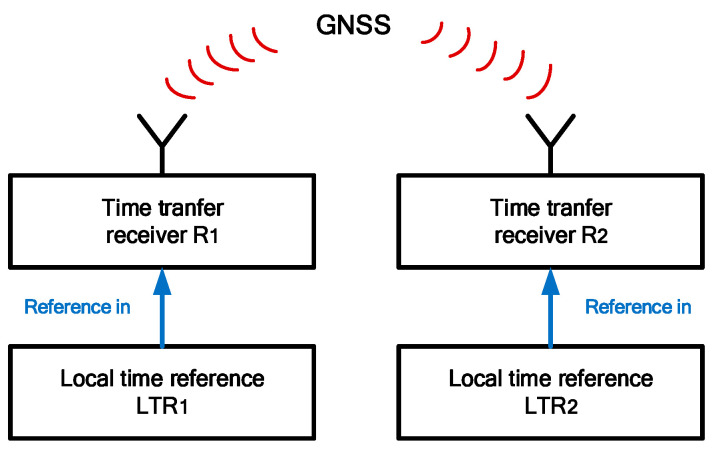
Experimental structure diagram.

**Figure 2 sensors-23-05328-f002:**
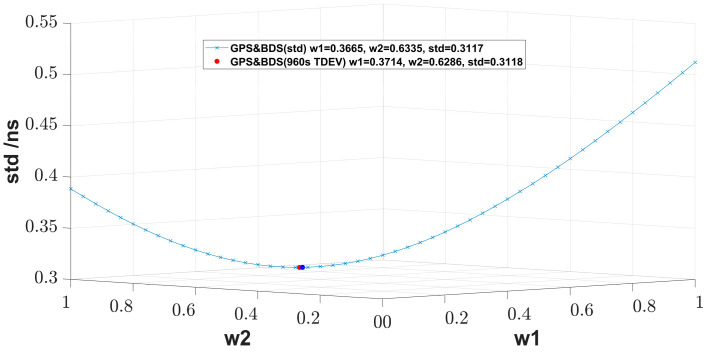
std for two-link weighted fusion with different weight allocation. The green dots show the enumeration of the weight combination with a step of 0.02.

**Figure 3 sensors-23-05328-f003:**
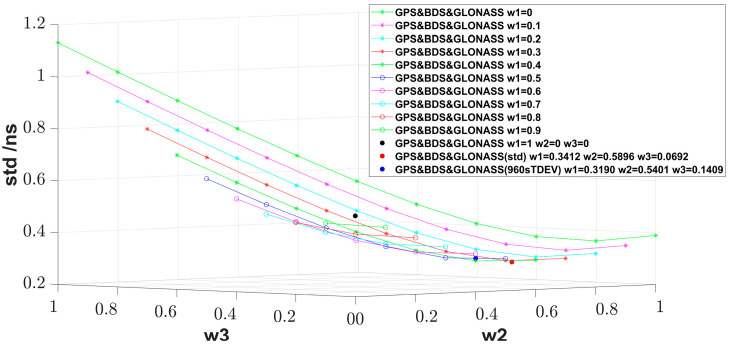
std for three-link weighted fusion with different weight allocation. The curves and the black dot show the enumeration of the weight combination with a step of 0.1.

**Figure 4 sensors-23-05328-f004:**
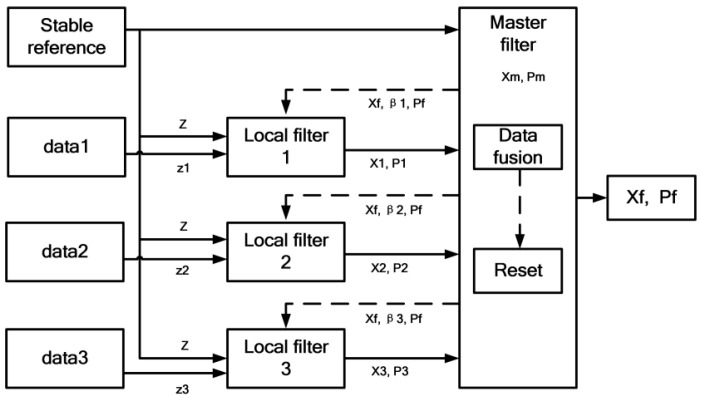
Scheme for Federated Kalman filter.

**Figure 5 sensors-23-05328-f005:**
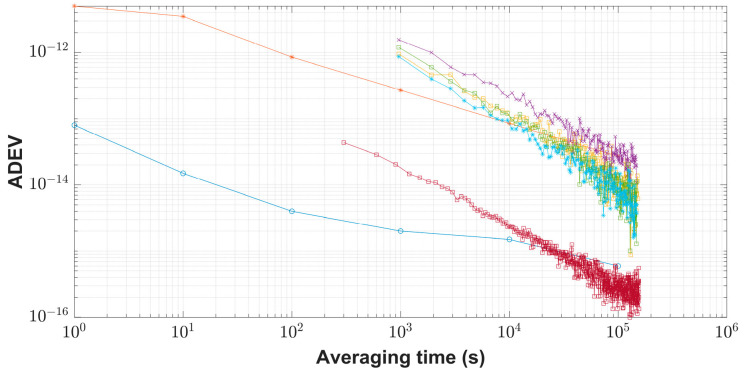
Relationship between the link performance and the clocks. Grey-blue circle (nominal specification for the frequency stability of the H-maser with low-phase noise, type VCH-1003M with option L), red asterisk (nominal specification for the frequency stability of the cesium clock with the high-performance tube, type Symmetricom 5071A), orange square (frequency stability of the CCD results via BDS B1I), purple cross (frequency stability via BDS L3B), green square (frequency stability via GPS C1), indigo asterisk (frequency stability via GPS P3), dark-red square (frequency stability via GPS CP). “CCD” represents the CCD results between two receivers with reference to UTC(NIM) (a time scale based on the H-maser, National Metrology Primary Standard of Atomic Time Scale) through GPS codes, BDS codes, or GPS CP measurements. For GPS CP processing, all the daily RINEX files were concatenated and NRCan_PPP software with IGS final products were used to obtain the solution of GPS CP measurements.

**Figure 6 sensors-23-05328-f006:**
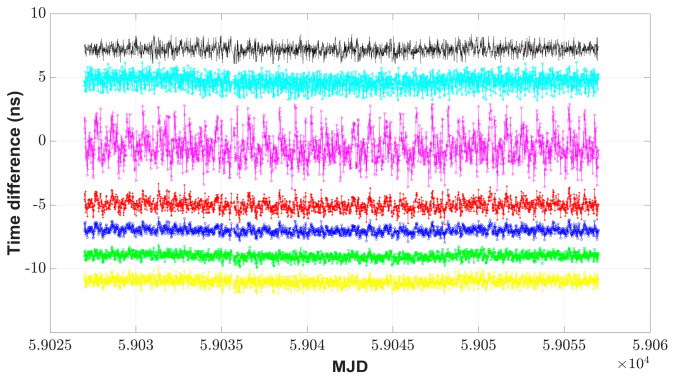
CCD results with use of multi-GNSS. The figure shows the results via BDS (black point), GPS (cyan Left-pointing triangle), GLONASS (magenta Right-pointing triangle), and the FKF link results for three GNSS systems (green square); the std-weighted method of different GNSS systems (yellow diamond); the unweighted average (red asterisk) of different GNSS systems and its results through the Kalman filter (blue circle). To facilitate viewing, the time difference curves are shifted up or down.

**Figure 7 sensors-23-05328-f007:**
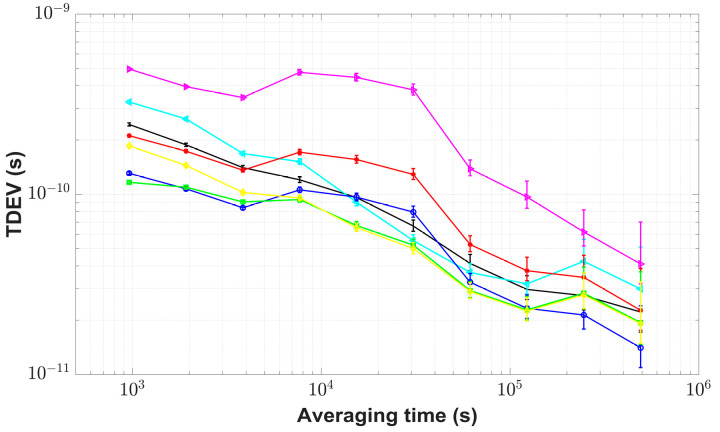
TDEV plots for CCD results with the use of multi-GNSS. The figure shows the results via BDS (black point), GPS (cyan Left-pointing triangle), GLONASS (magenta Right-pointing triangle), and the FKF link results for three GNSS systems (green square); the std-weighted method of different GNSS systems (yellow diamond); the unweighted average (red asterisk) of different GNSS systems and its results through the Kalman filter (blue circle).

**Figure 8 sensors-23-05328-f008:**
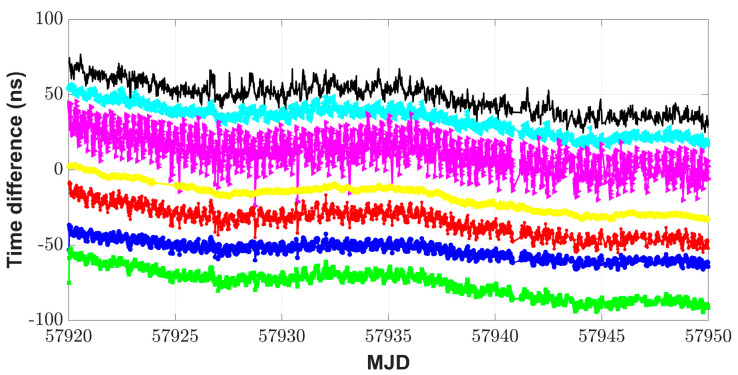
Remote time transfer results with use of multi-GNSS. The figure shows the results via BDS (black point), GPS (cyan Left-pointing triangle), GLONASS (magenta Right-pointing triangle), GPS CP (yellow diamond), and the FKF link results for three GNSS systems (green square); the unweighted average method (red asterisk) and its results through the Kalman filter (blue circle). To facilitate observation, the time difference curves are shifted up or down.

**Figure 9 sensors-23-05328-f009:**
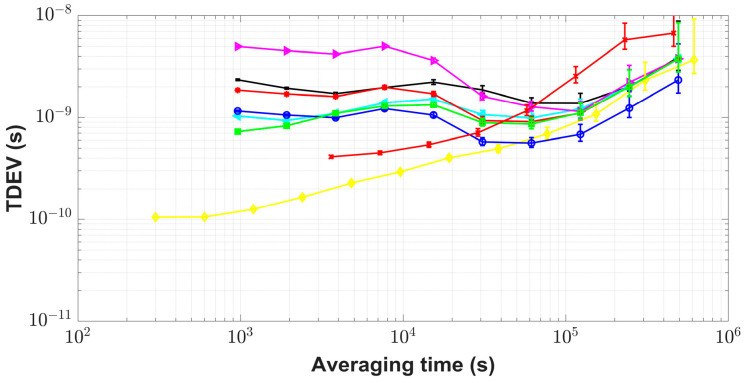
TDEV plots for remote time transfer results with use of multi-GNSS. The figure shows the results via BDS (black point), GPS (cyan Left-pointing triangle), GLONASS (magenta Right-pointing triangle), GPS CP (yellow diamond), and FKF link results for three GNSS systems (green square); the unweighted average method (red asterisk) and its results through the Kalman filter (blue circle). The cesium clock specification (red cross) was acquired from the measurement of one cesium clock referenced to UTC(NIM).

**Table 1 sensors-23-05328-t001:** std of two-link weighted fusion with different weight allocation.

Weight Allocation	GPS	BDS	std
	*w* _1_	*w* _2_	
two GNSS systems	0.0000	1.0000	0.3885
	0.1000	0.9000	0.3542
	0.2000	0.8000	0.3287
	0.3000	0.7000	0.3143
	0.4000	0.6000	0.3126
	0.5000	0.5000	0.3237
	0.6000	0.4000	0.3464
	0.7000	0.3000	0.3786
	0.8000	0.2000	0.4181

**Table 2 sensors-23-05328-t002:** std of three-link weighted fusion with different weight allocation.

Weight Allocation	GPS	BDS	GLONASS	
	*w* _1_	*w* _2_	*w* _3_	std
three GNSS systems	0.0000	0.0000	1.0000	1.1314
	0.0000	0.2000	0.8000	0.9085
	0.0000	0.4000	0.6000	0.6965
	0.0000	0.6000	0.4000	0.5093
	0.0000	0.8000	0.2000	0.3850
	0.0000	1.0000	0.0000	0.3892
	0.2000	0.0000	0.8000	0.9155
	0.2000	0.2000	0.6000	0.6957
	0.2000	0.4000	0.4000	0.4942
	0.2000	0.6000	0.2000	0.3452
	0.2000	0.8000	0.0000	0.3294
	0.4000	0.0000	0.6000	0.7179
	0.4000	0.2000	0.4000	0.5116
	0.4000	0.4000	0.2000	0.3501
	0.4000	0.6000	0.0000	0.3131
	0.6000	0.000	0.4000	0.5583
	0.6000	0.2000	0.2000	0.3983
	0.6000	0.4000	0.0000	0.3465
	0.8000	0.0000	0.2000	0.4766
	0.8000	0.2000	0.0000	0.4179
	1.0000	0.0000	0.0000	0.5117
weight using std	0.3412	0.5896	0.0692	0.3031
weight using TDEV	0.3190	0.5401	0.1409	0.3158

## Data Availability

Data sharing not applicable.

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
