# Peer review of "A Multi-Global Navigation Satellite System (GNSS) Time Transfer Method with Federated Kalman Filter (FKF)"

_sensors, 2023, doi:10.3390/s23115328_

Round 1

Reviewer 1 Report

This manuscript investigated a multi-GNSS time transfer method based on fusion of multi-GNSS measurement with different weights using federated Kalman filter. The data fusion effects among the enumerated different weights, standard deviation allocated weights and time deviation (TDEV) allocated weights have been compared.

1) In Tables 1 and 3, how to get the values of “std”? What is the unit of “std”?

2) Why “std allocated weight” scenario was selected? with theory deduction or experimental result?

3) References from the latest research should be cited.

The Originality / Novelty of this manuscript is not so strong.

Author Response

Ⅰ.Comments and Suggestions from reviewer 1:

The present work presents the results of an application concerning the studies on the effects of the different weight allocation for multi-measurements of GNSS time transfer with particular reference to the Federation Kalman Filter.

(1) The work appears complete and well structured both in terms of presentations and in terms of applied results produced (perhaps the introduction should be improved). However, in order to better understand the proposed methodology, in the opinion of the writer, it is necessary to revisit the subparagraph "2.3. Multi-GNSS time transfer" by proceeding to a detailed description of the flowchart shown in Figure 4 and of the sentence reported on line 169 which refers to it.

Thank you very much for your careful review of our manuscript. We have added a detailed description of the flowchart shown in Figure 4 as " could be allocated with std weighted method. At the same time, the local filtering is implemented on each measurement input. In Figure 4, data1, data2, and data3 represent the results of BDS B1I, GPS C1 and GLONASS C1 code measurements in CCD experiments, or the results of BDS L3B, GPS P3 and GLONASS P3 code measurements in remote time transfer experiments. ".

In page 6, line 170, We have revised as “These results are fed back to the local filter to correct the system itself, so that the state of the Kalman filter in the process tends to be zero and then the closed loop of the federated Kalman filter is realized.”

(2) Furthermore, it should be better clarified if the proposed algorithms have been implemented on their own or if other software has been used for data processing other than the one reported: NRCan_PPP software.

Thank you very much for your careful review of our manuscript. We used the FKF algorithm to complete the fusion of CCD results of different GNSS time links. NRCan_PPP was used to obtain the solution of GPS CP measurements, not data fusion. Supplementary notes had been added.

(3) It would also be advisable to insert a section where the proposed results can be discussed and where perhaps it would be better to highlight the effects of the results obtained even on baselines of different lengths.

We are grateful for the suggestion. We have added a discussion section as " CCD and the long baseline with 7656 km length experiments characterized the typical performance of GNSS time transfer links, such as noise level and accuracy. The results showed that the additional time stability of FKF link for three GNSS systems were better than the single GNSS system with code measurements. Longer baselines introduce more link noise. Therefore, in principle, FKF time link can achieve similar performance when the baseline length is shorter than 7656 km.".

(4) The bibliography should be updated and increased by also taking into consideration other sector studies.

Thank you for your valuable consideration. The references [9-11] are added from different universities and institutes.

Reviewer 2 Report

The article deals with Global Navigation Satellite System (GNSS) measurements and optimization of the method of information transfer. Thus, the article meets the goals of the journal Sensors. In order for the article to be published, the authors must make some improvements to the manuscript.

 1.      A title in which two of the six words are abbreviations sounds unfortunate. I suggest that the full name of the Global Navigation Satellite System (GNSS) and Federated Kalman Filter (FKF) be given in the title of the article.

2.      The list of references is very poor. In addition, of the 16 references, 4 belong to the main author. It is necessary to give a more complete literature review.

3.      2.1 GNSS time transfer. More information should be given regarding Common Clock Difference (CCD) experiments. The works of Liang et al. (2017, 2018, 2019) are difficult to find from the information given in the references. In addition, only Liang et al., 2018 is listed. The rest of the publications contain a full list of co-authors.

4.      The Federated Kalman Filter (FKF) should be described more completely.

5.      Fig. 4, 6. Explain the abbreviation ADEV; Fig. 5, 7. Explain the abbreviation MJD. Please check the entire manuscript to ensure that all abbreviations are explained the first time they are mentioned.

6.      Explain what TDEV plots include.

7.      The conclusions should be rewritten to make them more specific. Replace phrases like was discussed, was designed, have been implemented, etc.

8.      How do the conclusions correlate with Jian Tang et al (Applied Science, 2022), in which the authors evaluated the time transfer performance with a BDGIM-Based Phase-Smoothed Pseudorange Algorithm for BDS-3 High-Precision Time Transfer.

Author Response

Ⅱ.Comments and Suggestions from reviewer 2:

This manuscript investigated a multi-GNSS time transfer method based on fusion of multi-GNSS measurement with different weights using federated Kalman filter. The data fusion effects among the enumerated different weights, standard deviation allocated weights and time deviation (TDEV)allocated weights have been compared.

(1) ln Tables 1 and 3, how to get the values of "std"? What is the unit of "std"?

Thank you very much for your careful review of our manuscript. The “std” in Table 1 and Table 3 is the standard deviation of the fusion of CCD results which come from GNSS time links during MJD 59027 - MJD 59056. The unit of "std" is nanosecond.

(2) Why "std allocated weight” scenario was selected? with theory deduction or experimental result?

Thank you very much for your careful review of our manuscript. "Std allocated weight” scenario was selected with experimental result. The std of the fusion of different CCD time links is used to evaluate the suitability of the three weight allocation methods. A smaller std indicates a more suitable weight allocation method. From lines 92 to lines 108 on page 3, Table 1, Table 2, Figure 2, and Figure 3, it can be seen that the std value of the results with std allocated weight is smallest. So "std allocated weight” was selected.

(3) References from the latest research should be cited.

Thank you for your suggestion. We have added references [9-11], which were published in 2022-2023.

Reviewer 3 Report

The present work presents the results of an application concerning the studies on the effects of the different weight allocation for multi-measurements of GNSS time transfer with particular reference to the Federation Kalman Filter.

The work appears complete and well structured both in terms of presentations and in terms of applied results produced (perhaps the introduction should be improved). However, in order to better understand the proposed methodology, in the opinion of the writer, it is necessary to revisit the subparagraph "2.3. Multi-GNSS time transfer" by proceeding to a detailed description of the flowchart shown in Figure 4 and of the sentence reported on line 169 which refers to it.

Furthermore, it should be better clarified if the proposed algorithms have been implemented on their own or if other software has been used for data processing other than the one reported: NRCan_PPP software.

It would also be advisable to insert a section where the proposed results can be discussed and where perhaps it would be better to highlight the effects of the results obtained even on baselines of different lengths.

The bibliography should be updated and increased by also taking into consideration other sector studies.

Author Response

Ⅲ.Comments and Suggestions from reviewer 3:

The article deals with Global Navigation Satellite System (GNSS) measurements and optimization of the method of information transfer. Thus, the article meets the goals of the journal Sensors. In order for the article to be published, the authors must make some improvements to the manuscript.

(1)  A title in which two of the six words are abbreviations sounds unfortunate. I suggest that the full name of the Global Navigation Satellite System (GNSS) and Federated Kalman Filter(FKF) be given in the title of the article.

Thank you for your great suggestion. We have replaced the two abbreviations in the title with the full name.

(2) The list of references is very poor. In addition, of the 16 references, 4 belong to the main author. lt is necessary to give a more complete literature review.

Thank you for your suggestion. We have added the references [9-11].

(3) 2.1 GNSS time transfer. More information should be given regarding Common Clock Difference (CCD) experiments. The works of Liang et al. (2017, 2018, 2019) are difficult to find from the information given in the references. ln addition, only Liang et al., 2018 is listed. The rest of the publications contain a full list of co-authors.

We are grateful for the suggestion. Information about CCD was added as “CCD experiments were used to evaluate the noise level between different satellites and receivers links(see Liang et al. (2017)) and verify the capacity of NIM-TF-GNSS-3 receiver and RinCGG software for BDS(see Liang et al. (2018)). The short-term stability of time links was evaluated by CCD experiments at NIM in Beijing(see Liang et al. (2019)).”. The full list of co-authors are supplement to reference [11].

(4) The Federated Kalman Filter (FKF) should be described more completely.

Thank you very much for your suggestions. We added a supplement to FKF at line 122 as “FKF is able to share information among data sources or sensors so as to improve the accuracy of estimates. Each sensor of FKF equips with its own Kalman filter, state and observed variables.”.

(5) Fig. 4, 6. Explain the abbreviation ADEV: Fig. 5, 7. Explain the abbreviation MJD. Please check the entire manuscript to ensure that all abbreviations are explained the first time they are mentionedy.

Thanks for your carefulness. We have checked and explained all abbreviations for the first time they are mentioned.

(6) Explain what TDEV plots include.

Thank you very much for your careful review of our manuscript. The meaning of each curve had been explained below the picture.

(7) The conclusions should be rewritten to make them more specific. Replace phrases like was discussed, was designed, have been implemented etc.

Thank you very much for your suggestions. We have revised the statement of conclusions. The conclusion is changed as " In this work, multi-GNSS measurements were used for time transfer. Three weight allocated methods were compared, and the std allocated weight method led to the best data fusion results. Compared to the best single system result, the std value for the results of two-link fusion and three-link with std allocated weight are improved by 19.8% and 22.0%, respectively. The FKF time transfer with std allocated weight for multiple GNSS systems implemented the time transfer experiments from multi-GNSS system. The FKF time links combinating the multi-GNSS measurement were generally improved in the stability at short terms compared to the original time link with the single GNSS measurements. In CCD experiments, the noise level (i.e., std) for the FKF time link was reduced by 43.2% and 35.9%, respectively, compared to the unweighted average method and the best single-GNSS measurements; the additional time stability for the FKF time link was improved as better than 117 ps, and the short-term time stability at 960 s was improved by 44.8% and 52.2%, respectively, compared to the unweighted average method and the best single-GNSS measurements. The long baseline results gave the similar effects. The results have shown the FKF time transfer for multi-GNSS measurements can bring effective improvements on the reliability and noise level.".

(8) How do the conclusions correlate with Jian Tang et al (Applied Science, 2022), in which the authors evaluated the time transfer performance with a BDGlM-Based Phase-Smoothed Pseudorange Algorithm for BDS-3 High-Precision Time Transfer.

Thank you for your comments. This paper focuses on multi-satellite system transmission, but in Jian Tang et al (Applied Science, 2022), the authors evaluated the time transfer performance with a BDGlM-Based Phase-Smoothed Pseudorange Algorithm for BDS-3 High-Precision Time Transfer, and it aims at Beidou satellite system.
